# Effects of Exercise Training on Peripheral Muscle Strength in Children and Adolescents with Cystic Fibrosis: A Meta-Analysis

**DOI:** 10.3390/healthcare10122520

**Published:** 2022-12-13

**Authors:** Anna Thorel, Margaux Machefert, Timothée Gillot, Francis-Edouard Gravier, Tristan Bonnevie, Pascal Le Roux, Clément Medrinal, Guillaume Prieur, Yann Combret

**Affiliations:** 1Rouen School of Physiotherapy, Rouen University Hospital, F-76000 Rouen, France; 2Physiotherapy Department, Le Havre Hospital, F-76600 Le Havre, France; 3Paris-Saclay University, UVSQ, Erphan, F-78000 Versailles, France; 4Normandie University, UNIROUEN, CETAPS EA3832, F-76000 Rouen, France; 5Normandie University, UNIROUEN, UR 3830 GRHVN, Institute for Research and Innovation in Biomedicine (IRIB), F-76000 Rouen, France; 6ADIR Association, Rouen University Hospital, F-76000 Rouen, France; 7Pediatric Department, Le Havre Hospital, F-76600 Le Havre, France

**Keywords:** cystic fibrosis, pediatrics, physical activity, respiratory disease, exercise

## Abstract

Background: Exercise training is a cornerstone of care for people with cystic fibrosis (pwCF); it improves exercise capacity and health-related physical fitness, but no meta-analysis has investigated its effects on muscle function in young pwCF. The objective of this meta-analysis was to assess the effects of exercise on peripheral muscle strength in young pwCF. Methods: An electronic search was conducted in four databases (Pubmed, Science Direct, CENTRAL, and PEDRO) from their inception to July 2022. Grey literature databases (OpenGrey, the European Respiratory Society, the American Thoracic Society, and the European Cystic Fibrosis Society) were also consulted. Randomized controlled trials comparing any type of exercise with standard care in young pwCF (5 to 19 years old) were included. Two authors independently selected the relevant studies, extracted the data, assessed the risk of bias (using the Rob2 tool), and rated the quality of the evidence. Results: Ten studies met the inclusion criteria, involving 359 pwCF. Exercise training improved both lower and upper limb muscle strength (SMD 1.67 (95%CI 0.80 to 2.53), I^2^ = 76%, *p* < 0.001 and SMD 1.30 (95%CI 0.66 to 1.93), I^2^ = 62%, *p* < 0.001, respectively). Improvements were also reported in muscle mass and maximal oxygen consumption. Results regarding physical activity levels were inconclusive. The overall risk of bias for the primary outcome was high. Conclusions: Exercise training may have a positive effect on peripheral muscle strength in young pwCF. The evidence quality is very low and the level of certainty is poor. There is a need for high-quality randomized controlled studies to confirm these results.

## 1. Introduction

Cystic fibrosis (CF) is a multisystemic life-limiting autosomal recessive genetic disease caused by a defect in the CF transmembrane conductance regulator (CFTR) gene [1]. People with CF (pwCF) suffer from respiratory and digestive symptoms, but also health-related physical fitness diminution related to muscle dysfunction and exercise intolerance [2]. PwCF could have lower peripheral muscle strength and mass compared to healthy controls right from childhood [3,4]. On the one hand, peripheral muscle dysfunction may be a secondary consequence of the disease, caused by muscle disuse responsible for progressive deconditioning [5,6,7]. Other factors are also well-known contributors of muscle dysfunction, such as nutritional status, inflammation, exacerbations, and the use of corticosteroids [4,8]. Peripheral muscle weakness in pwCF could also come from a reduced muscle mass, since the difference in muscle strength between pwCF and healthy individuals seem to disappear when adjusting for muscle size [4,9,10]. On the other hand, recent data assume that intrinsic factors may contribute to muscle dysfunction, as the CFTR protein has been identified in the skeletal muscle and the neural system [11,12]. Furthermore, cellular mitochondrial dysfunction may be involved with the higher intra-cellular mitochondrial calcium rates recorded, which could consequently lead to impaired calcium salting-out during muscular contraction [13,14].

Therapeutic interventions improving muscle function and performance are therefore warranted. Exercise training is defined as a subset of physical activity (PA) that is planned, structured, and repetitive and aims to improve or maintain physical fitness [15]. Exercise is known to enhance health-related physical fitness, maximal aerobic capacity, and health-related quality of life (HRQOL) in pwCF [16]. However, whether exercise training will directly improve peripheral muscle strength in pwCF remains to be demonstrated—especially in the pediatric population. This meta-analysis was carried out to investigate the effects of exercise on peripheral muscle strength in young pwCF.

## 2. Methods

### 2.1. Study Registration and Methodology

This meta-analysis was designed on the basis of the Cochrane Handbook for Systematic Reviews of Interventions, and is reported according to the Preferred Reporting Items for Systematic Review (PRISMA) 2020 guidelines [17,18]. The protocol for the literature search was prospectively registered on the International Prospective Register for Systematic Reviews (PROSPERO: CRD42021228931).

#### 2.1.1. Data Search Strategy

An electronic search of the following databases: Pubmed, Science Direct, CENTRAL, and PEDro was carried out from the date of their inception to July 2022. The grey literature was also investigated through OpenGrey, and the conference abstracts of the American Thoracic Society, European Respiratory Society, and European Cystic Fibrosis Society annual congresses. References from the included studies were also screened. A sensitivity-maximising approach was carried out; the full search strategy is displayed in Appendix A.

#### 2.1.2. Eligibility Criteria

This meta-analysis focused on children (≥5 years old) and adolescents with CF (≤19 years old), according to the World Health Organization (WHO) classification [19]. The inclusion criteria were: (1) studies that included children and adolescents with a confirmed diagnosis of CF; (2) randomized controlled trials (RCTs) and cross-over randomized trials (including conference papers) that assessed one or more of the considered outcomes; (3) studies published in English and in French. Full-text formats were only taken into account when studies were available as full texts and conference abstracts. Other study designs and studies including both children and adults with CF were not included.

#### 2.1.3. Type of Intervention and Control

Exercise training was considered to be any exercise training program. The control arm had to receive standard medical and physiotherapy treatments, without any additional exercise intervention.

#### 2.1.4. Type of Outcome Measures

The primary outcome was peripheral muscle strength (i.e., lower limb muscle (LLM) and upper limb muscle (ULM) strength), regardless of the measurement procedures employed. Static measurements (i.e., fixed dynamometry), dynamic measurements (i.e., isokinetic dynamometry, one-repetition maximum, five-repetition maximum), or functional measurements (i.e., horizontal jump test, sit-to-stand test) were considered.

Secondary outcomes were muscle mass, maximal aerobic capacity, and PA levels. Maximal aerobic capacity was restricted to the measurement of VO_2_peak using cardio-pulmonary exercise testing (CPET).

### 2.2. Study Selection and Data Collection Process

The research was conducted by the two main authors (AT, YC), imported to a Microsoft Excel file with duplicates deleted. All titles and abstracts were independently screened manually (AT, YC). If necessary, a third reviewer (GP) was consulted in case of disagreements.

Data extraction was independently performed by two authors (AT, YC). If necessary, a third reviewer (GP) was called upon to resolve any disagreements. The following data were extracted: study details (authors, publication date); sample characteristics (sample size, age, eligibility criteria, clinical status, and mean forced expiratory volume in one second (FEV1)); interventions (type and duration, frequency, intensity); assessment procedures, and results. Mean (standard deviation or confidence interval) or median (interquartile range) change values were extracted. For outcomes expressed as a figure, data were extracted using https://automeris.io/WebPlotDigitizer/, accessed on 22 November 2022. Corresponding authors were contacted if meaningful data were missing from any study. Results were described narratively whenever data were not available for meta-analysis.

### 2.3. Assessment of Risk of Bias in the Included Studies

Two authors (AT, YC) independently evaluated the methodological quality of the studies, for each of the outcomes, using the risk of bias 2 (RoB2) tool described in the Cochrane Handbook for Systematic Reviews of Interventions [20]. The methodological criteria were: (i) randomization process; (ii) deviations from intended interventions; (iii) missing outcome data; (iv) measurement of the outcome; (v) selection of the reported results, and (vi) overall risk of bias.

### 2.4. Data Analysis

The meta-analysis was performed using the general approach for continuous outcomes described for parallel-group RCT using the Cochrane Collaboration’s Review Manager Software RevMan. Since studies revealed pre-treatment differences in muscle strength, data were imputed as mean changes between pre- and post-treatment. Mean changes in the intervention and control groups and their standard deviations, along with the sample size, were included in the software. In the case of data expressed as median and interquartile range when the data distribution was skewed, the data were transformed using the method described by Wan et al. [21]. If not directly available, mean changes were calculated on the basis of: (i) the mean values for each condition or mean between-condition differences, (ii) the standard deviation or confidence interval by imputing the standard error, (iii) the *p*-value, using a conservative approach when the *p*-value was given in the form *p* < 0.05; the retained *p*-value was *p* = 0.05, *p* < 0.01 (=0.01), or *p* < 0.001 (=0.001)), or (iv) a graph of measurements from which individual data for each condition could be extracted. In the case of multiple intervention groups proposing different exercise training programs (e.g., aerobic, resistance, anaerobic training), data from both training groups were compiled together. Standardized mean differences (SMD) were used when studies used different measurement tools or reported the results in different units for the same outcome [22].

The meta-analysis was performed using a fixed effect when the heterogeneity was low (I^2^ < 50%) and a random effect when the heterogeneity was moderate to high (I^2^ ≥ 50%). The risk of publication bias was assessed using funnel plots and Egger’s bias statistics for each outcome. A summary of the findings and the quality of evidence was undertaken using the Grading of Recommendations Assessment, Development, and Evaluation (GRADE) system by the two main authors (AT and YC) [23].

## 3. Results

### 3.1. Compliance with the Protocol Registration

Exercise training was considered rather than the broad concept of PA to focus our research on the particular effects of structured and planned exercise programs on muscle strength in young pwCF.

### 3.2. Selection and Characteristics of the Studies

A total of 4684 records were identified, among which 10 RCT were finally included, involving 359 pwCF aged 5 to 18 years old (Figure 1) [24,25,26,27,28,29,30,31,32,33]. Eight studies were available as full-text publications [24,25,27,28,30,31,32,33] and two as conference abstracts [26,29]. Studies that did not met the inclusion criteria and the reasons for their exclusion are listed in Appendix A. The agreement between AT and YC during the selection of the studies to be included was almost perfect, with a Cohen’s Kappa of 0.93 and a percentage of agreement of 97.4%. Funnel plots and Egger’s bias statistics did not suggest unequivocal publication bias (Appendix A).

The characteristics of the included studies are described in Table 1. Exercise training was conducted for 1 week to 24 months, with a frequency ranging from one to five times per week. The studies included 11 to 71 pwCF, aged 5 to 18 years old, with a wide range of bronchial obstruction severity (FEV1 ranging from 57.4 to 99.5%PV (predicted value)). PwCF were clinically stable in nine studies [25,26,27,28,29,30,31,32,33], and hospitalized for exacerbation in one study [24]. All the exercise protocols are described in Appendix A.

### 3.3. Risk of Bias

The individual risk of bias of the studies investigating the primary outcome is presented in Figure 2 and Figure 3. Appendix A presents a summary of the risk of bias for the primary outcome. Six out of the seven studies evaluating LLM and ULM strength had a high risk of bias. The main methodological issues identified were: caregivers and people delivering the interventions were aware of the participants’ assigned intervention; significant between-group differences at baseline for important outcomes (age, FEV1, BMI, muscle strength, or VO_2_peak); per-protocol analysis rather than intention-to-treat; flowchart unavailable; study protocol not recorded or unavailable; and selective outcome reporting. The risk of bias for the secondary outcomes is reported in Appendix A.

### 3.4. Effect of the Intervention

#### 3.4.1. Primary Outcome: LLM and ULM Strength

Eight studies investigated the effects of exercise training on peripheral muscle strength [24,25,26,27,28,30,32,33]. The methods for measuring muscle strength were heterogenous among studies. Measurement procedures are fully detailed in Appendix A. One study reported on only LLM strength while two studies reported on only ULM [24,32,33]. Another study measured peripheral muscle strength as the total maximal muscle strength of four muscle groups [25]. In the remaining studies, muscle strength was separated for ULM and LLM strength [26,27,28,30].

The meta-analysis for LLM strength (five out of eight studies, 137 pwCF) is shown in Figure 2. Exercise training significantly increased LLM strength, with a high estimated effect (SMD 1.67 (95%CI 0.80 to 2.53), I^2^ = 76%, *p* < 0.001). The meta-analysis for ULM strength (six out of eight studies, 139 pwCF) is shown in Figure 3. Exercise training significantly increased ULM strength, with an uncertainty ranging from a moderate to a high estimated effect (SMD 1.30 (95%CI 0.66 to 1.93), I^2^ = 62%, *p* < 0.001) (Table 2). One study did not report detailed results for muscle strength but stated that the exercise training program undertaken did not increased muscle strength [25]. Two studies did not perform between-group comparisons on muscle strength and reported solely within-group positive effects [24,26].

#### 3.4.2. Secondary Outcomes

Four studies reported results on fat-free mass (FFM). Three used measurements of skinfold thickness [24,27,28], and one used bio-electric impedance analysis [25]. In the meta-analysis (three out of four studies, 86 pwCF), exercise training showed a possible effect on muscle mass, with an uncertainty ranging from no effect to a high estimated effect (SMD 1.33 (95%CI 0.02 to 2.64), I^2^ = 85%, *p* = 0.05; Appendix A). One study did not report detailed results for muscle mass but stated that the exercise program did not increase muscle mass [25].

Nine studies reported results on VO_2_peak. All the studies measured VO_2_peak using a cardio-pulmonary exercise testing (CPET). Six used a treadmill protocol [24,27,28,31,32,33] and two used a cycloergometer protocol [25,26]. The remaining study did not specify the protocol [29]. In the meta-analysis (eight out of nine studies, 275 pwCF), exercise training significantly increased VO_2_peak, with a high estimated effect (MD 3.60 (95%CI 1.74 to 5.47), I^2^ = 59%, *p* < 0.001; Appendix A). One study measured VO_2_peak as a percentage of the predicted value and did not reveal intra-group differences [26].

Three studies reported results on PA levels. Two studies used the habitual activity estimation scale [25,31] and one study used an accelerometer with a 7-day diary [24]. In the meta-analysis (two out of the three studies, 83 pwCF), exercise training showed little to no effect on PA levels, with an uncertainty ranging from no effect to a high estimated effect (SMD 0.40 (95%CI −0.03 to 0.84), I^2^ = 0%, *p* = 0.07; Appendix A). One study did not report detailed results for PA levels but stated that the program undertaken did not increase habitual PA [25].

## 4. Discussion

The results of our systematic review can be summarized as follows: (1) exercise training seems to have a worthwhile benefit on (i) ULM and LLM strength and (ii) muscle mass in young pwCF; (2) exercise training could improve VO_2_peak with a clinically worthwhile benefit; and (3) uncertainty is high regarding whether exercise could increase PA levels in young pwCF.

Peripheral muscle strength could be improved using an exercise training protocol in young pwCF. The effect size was large, thereby suggesting a clinically meaningful benefit in this population. Exercise protocols were highly heterogenous, including different durations, frequencies, and intensities. Furthermore, different exercise training strategies were employed (i.e., resistance, aerobic, or anaerobic training), separated or altogether, that could induce different adaptations. Indeed, aerobic training increases endurance capacity, whereas resistance training would preferentially improve muscle strength [34]. For instance, one of the included studies, conducted by Selvadurai et al., compared both modalities and retrieved this exact tendency in young pwCF hospitalized for an acute exacerbation [24]. In clinical practice, these approaches are complementary and are usually performed together to maximize the results of exercise training. The present meta-analysis did not mean to compare these strategies. However, we observed that the studies that reported the highest benefits used this combination of aerobic and strength training. Unsurprisingly, this is in total agreement with the latest guidelines on exercise training for young pwCF, which propose at least three sessions of aerobic training along with two to three sessions of core and limb muscle strengthening per week for this population [35].

Measurements of peripheral muscle strength were extremely heterogenous among the studies included. This is of particular importance as these methods could have influenced the results retrieved from these studies. Hence, two studies used a 5RM protocol on the paediatric training machines that they used for resistance training [27,28]. There is here a possibly important confounding factor, since the training group could have benefited from a helpful learning effect compared to the control group, who did not train on these machines. For instance, it was previously reported that the maximal inspiratory pressure (i.e., a measurement of respiratory muscle strength) measured in pwCF could increase by 163% following an habituation protocol [36]. If the same mechanism applies for peripheral muscle strength, this could have artificially increased the between-group differences in these studies. This issue is central for pwCF as, to our knowledge, none of the measurement protocols employed in the included studies (except for isokinetic dynamometry in the study by Selvadurai et al.) revealed satisfactory clinometric properties in this population [37]. Among the methodological weaknesses that were noted, the measurement of the outcome was rated as having a high risk of bias for more than half of the studies included. It is still necessary in the years to come to determine valid and reliable tools of measuring muscle strength in young pwCF, at least for research purposes, and to determine for which individuals these measures will add relevant information for use in clinical practice [38].

One of the questions that was not directly addressed by our meta-analysis was to determine which pwCF will benefit from these exercise training protocols. The PwCF samples in the studies included were heterogenous, thereby explaining the high heterogeneity in the meta-analysis. Not all participants presented with muscle dysfunction or reduced exercise capacity, and so were not selected on the basis of these criteria. Studies have shown in the past a trend towards a reduced lower limb muscle strength, mass, and endurance in children with CF compared to healthy peers [39,40]. Intuitively, supervised exercise training will be suitable for these young pwCF with muscle dysfunction, but it is far from being the case for this population as a whole. One problem here is that the majority of the studies included did not express baseline muscle strength compared to predicted values. Respiratory status (i.e., FEV1) was heterogeneous across studies, and the same could be true for muscle strength since both parameters are correlated in this population [41]. Nevertheless, it is clear that not all young pwCF—especially those with preserved muscle function—will require specific, periodised, supervised training [42]. Rather, one of the major issues for these children is to determine the means by which they can engage in long-term PA and sport. Regarding this aspect, our meta-analysis did not show an increase in the daily amount of PA in these children. However, only three studies reported on this point [24,25,31]. Additionally, it is likely that the assessment of this criterion immediately at the end of a training programme is not appropriate timing. The end of a periodized and intense programme in the short-term may be synonymous with fatigue, or even lassitude, and this criterion should rather be assessed in the medium- or long-term. A recent systematic review including children and adult pwCF reported an increase in PA levels in 11 out of 15 studies following an exercise training programme including activity counselling [43]. In addition, two studies in our analysis showed that the benefits gained during training tended to be lost as early as 4 weeks after the end of the programme, without a strategy for maintaining the gains [27,28]. In contrast, the study by Del Corral et al., which used a video game training programme carried out at home, showed almost total maintenance 1 year after the initial 6-week programme [30]. This suggests that these programmes will achieve short-term benefits that may be highly relevant for some sub-groups of pwCF, but that other strategies (e.g., a motivational intervention) are needed to promote PA and sport. Amongst others, facilitating factors and barriers to PA, self-efficacy, or competing priorities should be explored and taken into account [44]. These aspects will become even more interesting in the current era of CFTR modulators, as the nutritional, respiratory, and physical status of young pwCF has never been so well preserved [45,46].

### 4.1. Limitations

This review has several limitations: The methods used to assess peripheral muscle strength were highly heterogenous, thereby limiting the external validity of the results obtained. Exercise training modalities were different across studies in terms of intensity, duration, and frequency. The overall quality of the evidence was very poor, mainly because of low sample sizes and selective outcome reporting, which could obviously limit the certainty of our results. Finally, the languages of the studies included were limited to English and French.

### 4.2. Perspectives

A major limitation of our meta-analysis is that inclusion criteria were wide-ranging in the studies. In the meantime, the muscle strength measurement procedures—as well as the exercise training protocols—were highly heterogenous. We suggest that future research aiming to investigate the particular value of a periodized exercise training protocol focus on pwCF with muscle dysfunction or reduced endurance capacity. This could reduce the heterogeneity between studies and led to more robust data compilation. As physical condition in pwCF is being increasingly preserved, it would not be appropriate to study the effect of these training programmes on the pwCF population as a whole. Additionally, muscle strength measurements should be standardized and widely described to enhance the replicability and external validity of findings. Exercise training procedures should comply with international guidelines and comprise both endurance and strength training. Finally, long-term engagement in PA should be a major aim and complementary strategies could be implemented during training programmes.

## 5. Conclusions

There is very low-quality evidence that exercise training improves peripheral muscle strength in young pwCF with a large estimated effect. Exercise training also seems to improve muscle mass and aerobic capacity, but failed to show positive results on PA levels in our analysis. Robust randomized controlled trials are warranted to strengthen these findings and increase the level of certainty regarding them.

## Figures and Tables

**Figure 1 healthcare-10-02520-f001:**
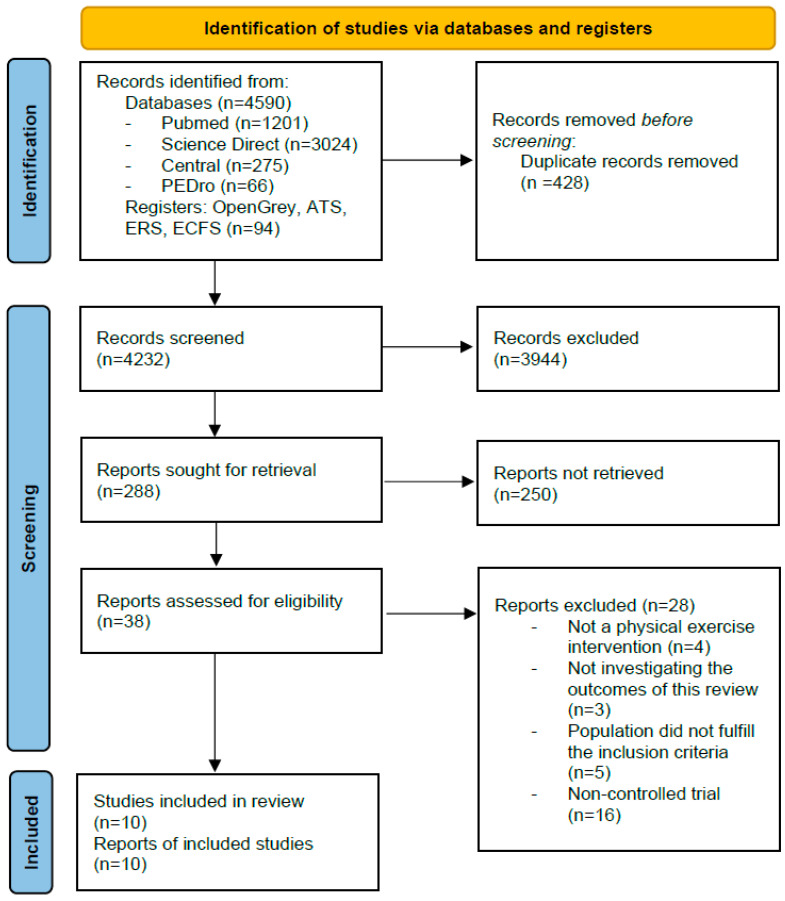
Flow diagram of the included studies.

**Figure 2 healthcare-10-02520-f002:**
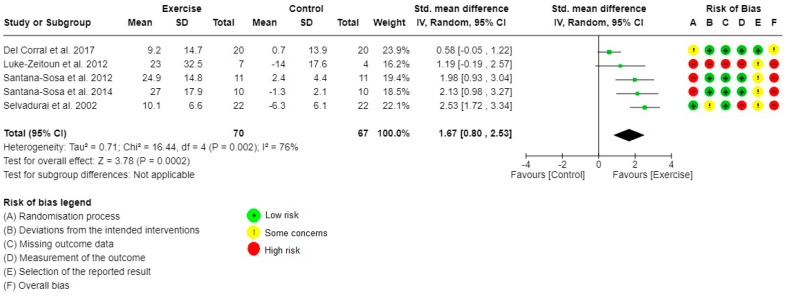
Forest plot for comparisons and risk of bias: Exercise versus Control, Outcome 1: Lower limb muscle strength.

**Figure 3 healthcare-10-02520-f003:**
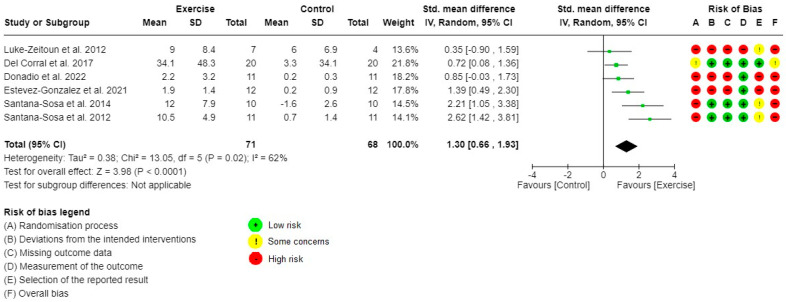
Forest plot for comparisons and risk of bias: Exercise versus Control, Outcome 1: Upper limb muscle strength.

**Table 1 healthcare-10-02520-t001:** Study details and results.

Study Details	N	Characteristics	Study Design	Interventions	Main Findings (Between-Group Comparisons Only)	
					Benefits of Exercise Training	Drawbacks or No Effects
Selvadurai et al., 2002 [24]	66	Exacerbation FEV1: 57.4 (±17.4)%PV Age: 13.2 (±2) yo Age range: 5–16	RCT, Inpatient, supervised, 1-week program, 5 times/week	I.1: aerobic exercise I.2: strength training (ULM and LLM) C: standard chest physiotherapy	-	LLM, FFM, VO_2_peak and PA level: not reported
Klijn et al., 2004 [25]	20	Stable FEV1: 78.7 (±19.9)%PV Age: 13.9 (±1.7) yo Age range: 9–18	RCT, Outpatient, supervised, 12-weeks program, 2 times/week	I: anaerobic exercise (ULM and LLM) C: habitual CF care	-	ULM and LLM strength, FFM, VO_2_peak, PA level: no difference (unreported result details)
Luke-Zeitoun et al., 2012 [26]	11	Stable FEV1: 99.5 (±12)%PV Age: 12 (±2) yo Age range: 8–16	RCT, Home-based, unsupervised, 6-months program, unknown frequency	I: individualized exercise program (not precise) C: habitual CF care	-	ULM, LLM and VO_2_peak: not reported
Santana Sosa et al., 2012 [27]	22	Stable FEV1: 1.8 (±0.2) L Age: 10.5 (±2.5) yo Age range: 5–15	RCT, Inpatient, supervised, 8-weeks program, 3 times/week	I: aerobic and strength exercise (ULM, LLM and core) C: habitual CF care + PA information	↗ULM (η^2^ = 0.44) and LLM (η^2^ = 0.50) strength ↗VO_2_peak (η^2^ = 0.15)	FFM: no difference (η^2^ = 0.11)
Santana Sosa et al., 2014 [28]	20	Stable FEV1: 1.6 (±0.2) L Age: 10.5 (±1) yo Age range: 6–17	RCT, Inpatient, supervised, 8-weeks program, 3 times/week	I: aerobic and strength exercise (ULM, LLM and core) + IMT C: Sham IMT + PA information	↗ULM (η^2^ = 0.72) and LLM (η^2^ = 0.62) strength ↗FFM (η^2^ = 0.34) ↗VO_2_peak (η^2^ = 0.51)	-
Ledger et al., 2016 [29]	71	Stable FEV1: 86.6 (±15.3)%PV Age: 10 (±3) yo Age range: 6–15	RCT, Inpatient, supervised, 24-months program, 1/week	I: aerobic and strength exercise (ULM, LLM and core) C: habitual CF care	-	VO_2_peak: no difference (+1.4 mL/min/kg (95%CI: −1.8 to 4.5))
Del Corral et al., 2018 [30]	40	Stable FEV1: 84.5 (±21)%PV Age: 11.8 (±3.2) yo Age range: 7–18	RCT, Home-based, supervised (virtual coach), 6-weeks program, 5 times/week	I: active video game exercise C: habitual CF care	↗ULM (MBT: 33.8 cm (95% CI 9.2 to 58.4; d = 1.2) and LLM (HJT: +9.2 cm (2.0 to 16.5; d = 1.2) strength	-
Gupta et al., 2019 [31]	52	Stable FEV1: 61.2 (±24.8)%PV Age: 12.5 (±3.1) yo Age range: 6–18	RCT, Home-based, unsupervised, 1-year program, 3 times/week	I: strength exercise (ULM and LLM) + vitamin D and calcium supplementation C: habitual PA + vitamin D and calcium supplementation	↗VO_2_peak (4.2 mL/min/kg (95%CI 1.2 to 7.1))	PA level: no difference (unreported result details)
Estevez-Gonzalez et al., 2021 [32]	24	Stable FEV1: −1.7 (±1.6) z-score Age: 12.3 (±3.3) yo Age range: 6–17	RCT, Inpatient, supervised, 8 weeks program, 3 times/week	I: strength exercise (ULM, LLM and core) C: habitual CF care	↗ULM (d = 1.6)	VO_2_peak: no difference (d = 0.24)
Donadio et al., 2022 [33]	33	Stable FEV1: -1.5 (±1.5) z-score Age: 12.6 (±3) yo Age range: 6-17	RCT, Inpatient, supervised, 8 weeks program, 3 times/week	I.1: strength exercise (ULM, LLM and core) I.2: strength exercise (ULM, LLM and core) + NMES (quadriceps and interscapular region) C: habitual CF care	↗ULM (η^2^ = 0.40)	VO_2_peak: no difference (unreported result details)

Main findings in the present table are limited to between-group differences. The results described in studies investigating within-group differences are described narratively in the main text. Abbreviations: C: control group; CF: cystic fibrosis; CI: confidence interval; d: Cohen’s d (effect size); FEV1: forced expiratory volume in one second; FFM: fat-free mass; HJT: horizontal jump test; I: intervention group; IMT: inspiratory muscle training; L: litres; LLM: lower limb muscles; MBT: medicine ball throw; NMES: neuromuscular electrical stimulation; PA: physical activity; pwCF: people with CF; RCT: randomized controlled trial; ULM: upper limb muscles; VO_2_peak: maximal aerobic capacity; yo: years old; %PV: percentage of the predicted value; η^2^: partial eta squared.

**Table 2 healthcare-10-02520-t002:** GRADE Summary of findings.

Exercise Training Compared to Controls for Young pwCF					
Outcomes	Anticipated Absolute Effects * (95% CI)		№ of Participants (Studies)	Certainty of the Evidence (GRADE)	Comments
	Risk with Control	Risk with Exercise Training			
1.1. Lower limb muscle strength	SMD 1.67 higher (0.80 higher to 2.53 higher)		137 (5 RCTs)	⨁◯◯◯ VERY LOW ^a,b,c,e^	Exercise training may have a positive effect on LLM strength but the evidence is very uncertain. Heterogeneity is very high (I^2^ = 76%), LLM strength measurements are heterogenous and risk of bias is high
1.2. Upper limb muscle strength	SMD 1.30 higher (0.66 higher to 1.93 higher)		139 (6 RCTs)	⨁◯◯◯ VERY LOW ^a,b,c,e^	Exercise training may have a positive effect on ULM strength but the evidence is very uncertain. Heterogeneity is high (I^2^ = 62%), ULM strength measurements are heterogenous and risk of bias is high
Muscle mass	SMD 1.33 higher (0.02 higher to 2.64 higher)		86 (3 RCTs)	⨁◯◯◯ VERY LOW ^a,b,c^	Exercise training may have little to no effect on muscle mass, and the evidence is very uncertain. Heterogeneity is very high (I^2^ = 85%), baseline differences prevent clear interpretation and risk of bias is high
VO_2_peak	MD 3.60 higher (1.74 higher to 5.47 higher)		275 (8 RCTs)	⨁⨁◯◯ LOW ^a,b,c^	Exercise training may have a positive effect on VO_2_peak but the evidence is uncertain. Heterogeneity is high (I^2^ = 59%), baseline differences were retrieved on maximal aerobic capacities and risk of bias is high
Physical activity level	SMD 0.40 higher (−0.03 lower to 0.84 higher)		86 (2 RCTs)	⨁◯◯◯ VERY LOW ^a,d,e^	Exercise training may have little to no effect on physical activity level, and the evidence is very uncertain. Risk of bias is high (self-reported physical activity level) and methods of measurement are heterogenous

* The risk in the intervention group (and its 95% confidence interval) is based on the assumed risk in the comparison group and the relative effect of the intervention (and its 95% CI). CI: Confidence interval; SMD: Standardised mean difference. GRADE Working Group grades of evidence. High certainty: We are very confident that the true effect lies close to that of the estimate of the effect. Moderate certainty: We are moderately confident in the effect estimate: The true effect is likely to be close to the estimate of the effect, but there is a possibility that it is substantially different. Low certainty: Our confidence in the effect estimate is limited: The true effect may be substantially different from the estimate of the effect. Very low certainty: We have very little confidence in the effect estimate: The true effect is likely to be substantially different from the estimate of effect. Explanations: ^a^. High risk of bias (lack of blinding for assessors and patients, lack of information on randomization process, statistical analysis often inappropriate). ^b^. High heterogeneity (I^2^ > 50%). ^c^. Some studies have a serious between-group baseline difference (baseline demographic characteristics values, peripheral muscle strength or VO2peak values). ^d^. High risk of bias due to the fact that the outcome assessment is made by the participants themselves (reported physical activity level). ^e^. Heterogeneity of outcome measurement protocol.

## Data Availability

Data will be available from the authors upon reasonable request.

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
