# Peer review of "Effects of Exercise Training on Peripheral Muscle Strength in Children and Adolescents with Cystic Fibrosis: A Meta-Analysis"

_healthcare, 2022, doi:10.3390/healthcare10122520_

Round 1

Reviewer 1 Report

This meta-analysis aims to assess the effects of exercise on peripheral muscle strength in young pwCF. This research is considered interesting and necessary in the field of healthy children and adolescents. However, several shortcomings are observed in the methodology and results section that cannot be overlooked. Authors are recommended to deepen their description.

It is not specifically indicated how the keywords, booleans, title search, abstract... were used and cannot be replicated. This aspect is realy relevant in order to reply the search.

Although it is indicated that two reviewers independently searched the databases, it is indicated how the synthesis was carried out or how to proceed if there were discrepancies; also it is indicate who and how data extraction is performed, however it is not included the Cohen's Kappa in relation to the inter-rater reliability for the two authors.

The analysis of the results is very detailed and the authors are congratulated. They also include the risk of bias, however Funnel plots, Egger bias statistics or Rosenthal's fail-safe are not included.

The authors present a large number of tables and figures showing the results of their study. These tables are very interesting and provide very valuable information, however, they present disparate formats, background colors in some cases and not in others, which make them difficult to understand. It is recommended that the authors unify these aspects of style.

Reviewer 2 Report

This is an interesting, complete and well designed study . 

Main findings  and limitations are well described .

Results are interesting and , in part , comply with the guidelines indications 

The problem of standardisation for strenght exercise has been adressed for other diseases and might pe properly defined. 

The continuity over time is the main limitation: a motivational intervention could help to improuve this aspect.

Reviewer 3 Report

Dear authors,

Congratulations for the work done. Let me suggest some changes to strengthen the work:

- If the authors use the PICO strategy to search the databases, in Appendix 1 they need to indicate section O "Outcome".

- I suggest authors change the keywords to others that are not contained in the title, in order to gain greater visibility in future searches.

- I suggest the authors improve the quality of the figures, to facilitate their reading.
